# Inference of glioblastoma migration and proliferation rates using single time-point images

Emil Rosén[1], Hitesh Bhagavanbhai Mangukiya [1], Ludmila Elfineh[1], Rebecka Stockgard[1], Cecilia Krona [1], Philip Gerlee[2,3] & Sven Nelander [1]✉

Cancer cell migration is a driving mechanism of invasion in solid malignant tumors. Anti-migratory treatments provide an alternative approach for managing disease progression. However, we currently lack scalable screening methods for identifying novel anti-migratory drugs. To this end, we develop a method that can estimate cell motility from single end-point images in vitro by estimating differences in the spatial distribution of cells and inferring proliferation and diffusion parameters using agent-based modeling and approximate Bayesian computation. To test the power of our method, we use it to investigate drug responses in a collection of 41 patient-derived glioblastoma cell cultures, identifying migration-associated pathways and drugs with potent anti-migratory effects. We validate our method and result in both in silico and in vitro using time-lapse imaging. Our proposed method applies to standard drug screen experiments, with no change needed, and emerges as a scalable approach to screen for anti-migratory drugs.

[1] Dept of Immunology, Genetics, and Pathology, Uppsala University, Uppsala, Sweden. [2] Mathematical Sciences, Chalmers University of Technology, Gothenburg, Sweden. [3] Mathematical Sciences, University of Gothenburg, Gothenburg, Sweden. ✉email: sven.nelander@igp.uu.se

Distant metastasis and local invasion are key features of solid tumors, jointly accounting for 90% of all cancer deaths[1–3]. In glioblastoma (GBM), the most common malignant brain tumor[4,5], cancer cells migrate away from the central tumor mass into healthy brain tissue[6–8]. Such diffusively growing cancer cells are out-of-reach for surgical therapy and can seed tumor growth in crucial structures, such as the brainstem[9]. Intriguing new evidence suggests that particular existing GBM therapies, radiation and VEGF blockers, can sometimes aggravate GBM invasion[10–12]. Accordingly, identifying new pharmacological inhibitors of GBM cell invasion is a central priority of brain tumor research, both as a tool to block tumor spread and to mitigate any invasion-promoting effects of other therapies.

Here, we describe an accelerated and broadly applicable method for detecting drug-induced changes in the motility of patient-derived cancer cells grown in adherent culture. Generally, scoring effects on motility is challenging for two reasons. First, motility assays are harder to carry out in practice compared to cell growth assays. In a large majority of screens reported so far, the effects of drug compounds on migration are measured using the scratch wound assay, the cheapest and most straightforward migration assay[13–16]. Second, motility assays tend to generate results that are open to interpretation since detected effects are confounded with proliferation, or technical factors such as inconsistencies applied when scratching the surface[13,16,17]. Certainly, in the particular case of patient-derived GBM cell cultures, it remains somewhat unclear if motility is a good proxy for in vivo invasion, which includes engagement with several different anatomical structures, such as white matter fibers, the vasculature, and neurons. Some of these problems can be solved by using specialized tools for wound making and measuring cell proliferation in a parallel experiment[17]. However, such a methodology requires additional experimental costs and fails to identify drugs that affect both cell proliferation and migration. Other migration assays, such as Boyden invasion assays and time-lapse microscopy, help solve these issues but are more expensive and complex[13].

To remedy this problem, we propose a radically different method to estimate cell line and treatment-specific migration of patient-derived cancer cells while jointly measuring cell growth. The method adds no additional experimental cost. It can be applied to already generated high-content drug screen data sets designed to measure cell proliferation. The only requirements are end-point images of adherent in vitro cell cultures and known experimental parameters. The estimation is made possible by measuring differences in the spatial distributions of cells due to their proliferative and migratory behavior. Our method leverages combines AI-guided image analysis and an agent-based simulation model to extract both the proliferation rate and diffusion constant from single end-point images only. Applying the method to a large sample of untreated and drug-treated patient-derived cultures (Fig. 1a), we find that in vitro motility of GBM cells is associated with astrocyte/outer radial glia-like differentiation. We search a library of 94 compounds and nominate drugs with consistent and subclass-specific effects on cell motility. We distribute the proposed method as open software.

## Results

### Effects of proliferation and migration can be observed in the spatial distribution of cells
A typical readout while screening for cell proliferation is an end-point image of adherent cells. Proliferation rates can be estimated by counting the number of cells in the image and comparing it with the cell density of the initial number of seeded cells. However, we also note that the spatial distribution of cells may differ between cells from different patients (Fig. 1c, f). In some cases, cells are uniformly distributed (Fig. 1f), while in others, cells are clustered together (Fig. 1c).

We constructed an agent-based model simulating cell proliferation and motility of adherent cell cultures. Cells are initially seeded uniformly in a simulated well, dividing at a rate $\alpha$, dying at a rate $\mu$, and moving according to Brownian motion with diffusion constant $D$ (Fig. 1b). Additionally, collision forces are applied to each cell such that cells are distanced at least two cell radii apart and do not leave the well. We then compare the end-point states of two such simulations with identical parameters except for the diffusion constant (Fig. 1d, g). Cells in the simulation with lower diffusion constant tend to be clustered together (Fig. 1d), while those with higher diffusion are uniformly distributed (Fig. 1g). The pair correlation function (PCF) measures the relative cell density as a function of distance from each cell. It can capture the differences in spatial cell distributions when applied to both the simulated and real data[18] (Fig. 1e, h). For cells with high motility, the PCF will be constant around one (Fig. 1h), while the PCF for low motility cells will have a peak at a distance of two cell radii (Fig. 1e). In both cases, the PCF is zero for distances less than two cell radii due to repulsive cell-cell forces at those distances.

### Joint estimation of migration and proliferation using single-end-point images
We sought to determine if it is possible to estimate cell migration using end-point images only by counting the number of cells, computing the PCF, and utilizing the agent-based model. Approximate Bayesian computation (ABC) can be used to estimate parameters in a stochastic computational model[19], such as an agent-based model. By creating simulations with randomly selected parameter values, we retain simulations sufficiently close to the real data. The distribution of the retained parameter values will then approximate the true posterior distribution of the model parameters.

As a first step, we generated simulated end-point states using the agent-based model to see if the method could estimate the parameters from data generated by itself. Proliferation rate, diffusion constant, and cell radius varied between simulations while other parameters were kept constant. We computed the PCF and number of cells at the end of the simulation and estimated the model parameters of 1000 simulations for each simulation. For each parameter posterior, we calculated a point estimate by taking the mode of the posterior distribution (Fig. 2a, Supplementary Fig. 1a). The correlation between estimated and real proliferation rates were high (R = 0.87, $p < 0.001$, $n = 1095$), with a lower error for more proliferative cells (Supplementary Fig. 1b). The correlation between the real and estimated diffusion constant was lower (Fig. 2b), but still high (R = 0.50, $p < 0.001$, $n = 1095$). We could also estimate the cell radius (R = 0.52, $p < 0.001$, $n = 1095$, Supplementary Fig. 1c). Similar to the proliferation rate estimates, the diffusion constant estimates were more accurate for cells with higher proliferation (Fig. 2c). Theoretically, if there is no proliferation, both stationary and cells with arbitrarily high diffusion constants will have the same spatial distribution. Thus, the ABC algorithm may accept unrealistically high diffusion constant estimates in cases with a low proliferation rate.

We noticed that there was a weak correlation between the estimated diffusion constants and proliferation rates (R = −0.16, $p < 0.001$, $n = 1095$), despite that all parameters were independent in the simulation. However, if we only included simulations with proliferation rate $\log_{10}\alpha > -5.9$ (doubling time < 9 days), the correlation disappeared (R = 0.009, $p = $ ns). Thus, for all further experiments, we only measure the diffusion constant for wells with sufficiently high proliferation ($\log_{10}\alpha > -5.9$). Similarly, we

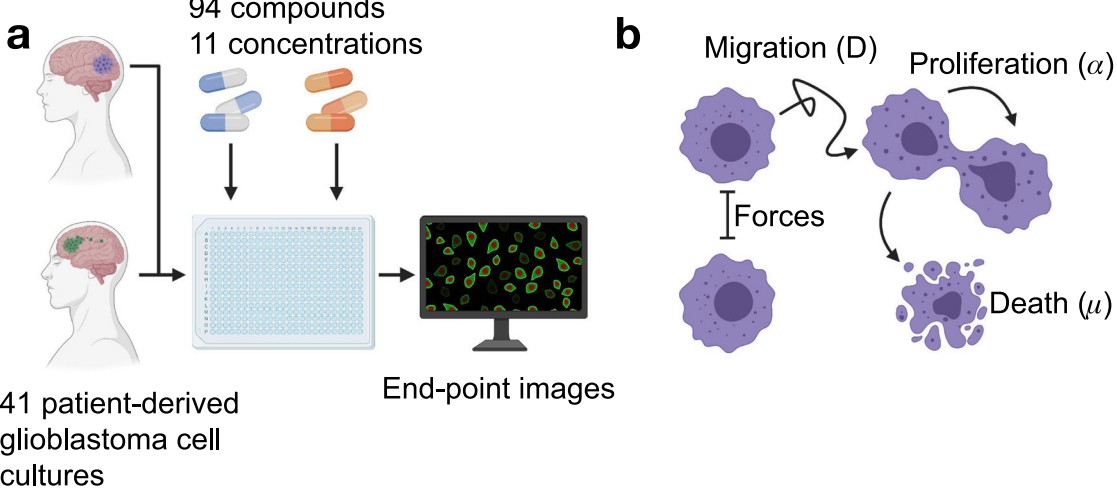

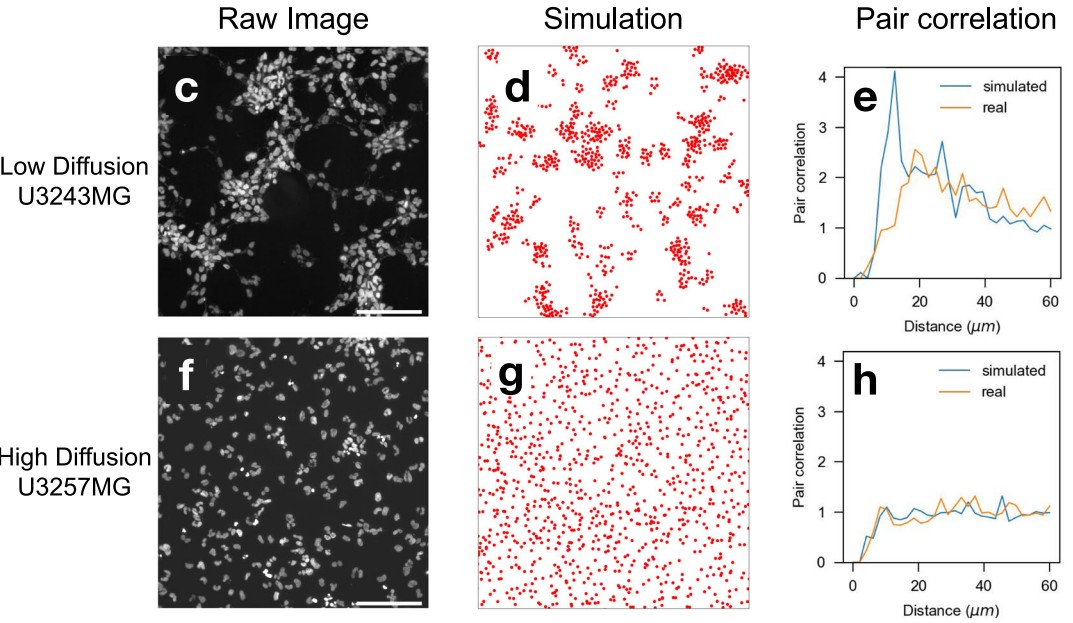

**Fig. 1 Endpoint imaging reveals different spatial cell distributions for low and highly migratory cells. a** Overview of glioblastoma drug screen experiment using patient-derived cell cultures. **b** Schematic of an individual agent-based cell migration model. **c**, **f** Endpoint image of two untreated cell cultures with different spatial distributions. **d**, **g** The end state of an individual agent-based simulation with equivalent proliferation but different diffusion. **e**, **h** Spatial cell distribution is captured in the pair correlation function. **c–e** Examples of low diffusion. **f–h** Examples of high diffusion. **a**, **b** contain clipared generated by Biorender.com. Scale bar is 100 μm.

found a weak ($R_s = -0.14$, $p < 0.05$, $n = 1095$, Supplementary Fig. 1D) correlation between cell radius and diffusion, indicating that cell radius induces a small bias on the diffusion estimates. However, the bias is relatively small, and we do not expect huge variations in cell radius.

As a next step, we aimed to see if using only end-point images would give similar estimates as those estimated from tracked cells from in vitro time-lapse data. We generated a time-lapse of phase-contrast images of adherent cells using an automated imaging system (IncuCyte S3). The cells were imaged in 96-well plates over 4 days with 1-h intervals between images. The experiments were repeated for five different initial cell densities, and ten different GBM patient-derived cell cultures (PDCs) from the Human Glioblastoma Cell Culture resource (HGCC)[20]. The cells in each image were segmented using a deep convolutional neural network (DCNN) and tracked over time (Supplementary

Fig. 2A–D). We calculated the proliferation rate by fitting an exponential function to the cell count in each frame. Additionally, using all tracked cells, we computed the mean square displacement (MSD) for each well, where the diffusion constant equals the slope of the MSD[21]. Of the 400 wells in the experiment, 54 wells were removed due to an insufficient proliferation rate. We compared the correlation between the model parameters estimated using all time points to those estimated from end-point images only. The diffusion constant estimates were more reliable for wells with a higher proliferation rate and initial cell density than wells with a low proliferation rate (Fig. 2d). As an additional requirement, we require the number of seeded cells > 125 (equivalent to 16 initial cells in the simulation) for accurate diffusion constant estimates, leaving 273 wells. Overall, the correlation between both the proliferation ($R_s = 0.51$, $p < 0.001$, $n = 273$, Supplementary Fig. 3a) and diffusion

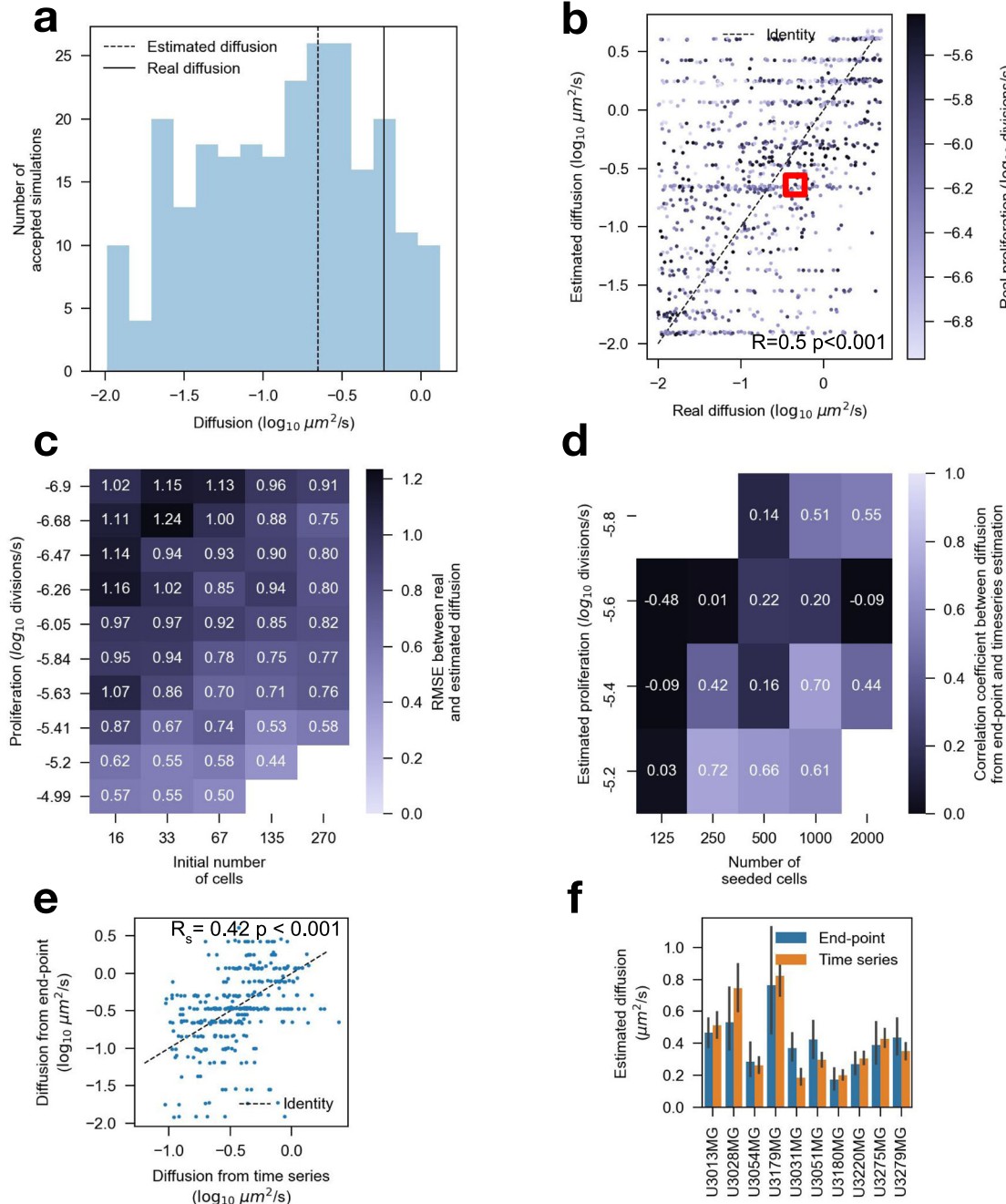

**Fig. 2 Approximate Bayesian computation (ABC) and individual agent-based model can identify low and highly diffusive cell populations from both simulated and real data using endpoint images. a** Example of the approximate diffusion posterior of the marked point in **b**. The solid and dotted line shows the real and estimated value, respectively. **b** Estimated diffusion parameters compared to real parameters from simulated data using 270 initial cells. Darker colored points have a higher proliferation rate. **c** RMSE between estimated and real diffusion parameters depends on the number of seeded cells and proliferation rate from simulation. **d** Spearman correlation between diffusion computed using endpoint images compared to diffusion estimates from tracked cells from the time-lapse validation experiment. Missing data in **c**, **d** due to early exit of simulations with improbable high cell count or if the number of wells in a bin was less than 10. **e** The diffusion constant was estimated from each well using tracked cells using time-lapse data compared with using an individual-based model and a single end-point image. **f** As in (**e**) but with wells aggregated over cell cultures; error bars as 95% confidence intervals. **a**, **c** $n = 1095$ simulations. (d-Ee $n = 346$ wells, $n \approx 20$ wells/bin. 54 wells with low proliferation were removed. **f** $n = 10$ cell cultures.

($R_s = 0.42$, $p < 0.001$, $n = 273$, Fig. 2e) estimates were high. The correlation coefficients were higher for both the proliferation ($R_s = 0.73$, $n = 10$, Supplementary Fig. 3b) and diffusion ($R_s = 0.83$, $n = 10$, Fig. 2f) when aggregating measurements over each PDC.

In a drug screen setting, treatments often induce cell death, which was not included in the above simulations. To investigate

how cell death might affect the method, we created an additional set of simulations with varying death rates ($\mu$ between 50–90% of $\alpha$) while omitting the death rate during estimation. The model estimated the net proliferation rate ($\alpha - \mu$) instead of the actual proliferation rate (Supplementary Fig. 1e). Although the added death rate induced a change to the PCF (Supplementary Fig. 1f), we noticed no correlation between the known death rate and the

estimated diffusion constant ($R_s = -0.08$, $p =$ ns, $n = 109$, Supplementary Fig. 1g). We further measured how the convergence rate of the diffusion constant was affected by death rate. While the estimates converged, we noted that the errors were consistently higher compared with simulations with no death rate (Methods, Supplementary Fig. 7e, g). Introducing cell death thus does not bias the diffusion estimates, but does make estimation more challenging.

Our results show that it is possible to estimate the diffusion constant from end-point images only, given that the net proliferation rate and number of cells in the image are sufficiently high. Although the method cannot directly estimate the death rate, it can measure treated cells' net proliferation rate and diffusion constant. We conclude that the method is suitable to screen for treatment effects on cell migration and proliferation.

**Estimating cell migration from a large cohort of glioblastoma cells.** After validating the method, we sought to apply our method to end-point images of a large-scale drug screen[22] (Fig. 1a). The screen was generated from a cohort of GBM PDCs from HGCC treated with small molecules[20]. The study consisted of three phases, iteratively reducing and selecting promising drugs, based on responses measured by the Alamar Blue assay while increasing the number of PDCs in each phase[20]. We opted to use phase 3 of the study due to superior image quality. As a first step, we aimed to characterize the 41 PDC-specific diffusion constant of untreated cells included in phase 3.

Similar to the previous experiment, we identified individual cells using a DCNN (Supplementary Fig. 2f–h) and estimated model parameters using ABC and the PCF for each well. Multiple sets of 100,000 simulations were created, each set with a different amount of seeded cells to match the experimental parameters used in the drug screen. To identify PDCs with acceptable diffusion constant estimates, we removed all PDCs where at least 50% of the untreated wells had insufficient proliferation rate ($log_{10}\alpha < -5.9$, $n = 94$ wells / PDC, Supplementary Fig. 4a, b). After filtering, 32 PDCs (78%) were retained. The mean of the $log_{10}$ estimates was used as a point estimate for each PDC.

The diffusion constant varied between the PDCs ($p < 0.001$, $n = 32$ PDCs, $n = 94$ wells/PDC, Fig. 3a), with some PDCs that were practically stationary. We did not, however, find an association between the PDC-specific proliferation rate and diffusion constant ($R_s = -0.04$, $p =$ ns, $n = 32$, Fig. 3b). Additionally, we found that the estimated cell radius varied around 5 μm Supplementary Fig. 4c, and were highly similar to the cell radius measured from the images ($R_s = 0.73$, $p < 0.001$, $n = 32$, Supplementary Fig. 4d. The method can thus infer cell nuclei size from the PCF alone. Additionally, we found a positive correlation between the measured cell radius and estimated diffusion ($R_s = 0.44$, $p < 0.05$, $n = 32$, Supplementary Fig. 4e). This correlation was not found in our simulations and is, therefore, unlikely to be due to the estimation procedure. The correlation between cell radius and diffusion is interesting and warrants further studies.

**Transcriptomic differences drive adherent cell migration.** HGCC includes gene expression data for 100 PDCs, including the 41 PDCs in the drug screen data set. We sought to understand if there were any transcriptomic differences between PDCs with high and low motility. We constructed a $p$-value histogram for the $p$ values computed between proliferation (Supplementary Fig. 5a) and diffusion (Supplementary Fig. 5b) with each gene. There was a clear enrichment for low $p$ values for the diffusion estimates. Next, we computed the three principal components of the gene expression data using all 100 PDCs. Spearman's

correlation coefficient was calculated between the proliferation rate and diffusion constant for each principal component. We found that the second principal component was significantly correlated with the diffusion constant ($padj < 0.05$, $n = 32$, Fig. 3c) with an $R^2 = 0.22$. Assuming that the second principal component represented a migratory signature, we sought to define it using all 100 PDCs. We used single-sample gene set enrichment analysis (SSGSEA) to get an enrichment score for each PDC and gene set of interest (Supplementary Data). Next, we computed the enrichment score between the invasive signature and gene sets (Fig. 3d, e).

We found a strong association ($padj < 10^{-14}$) between the classical subtype and migration as well as an association with astrocyte and radial glia signatures (Fig. 3d, Supplementary Fig. 5c). The mesenchymal subtype, by contrast, was not particularly associated with the diffusion constant. Matching to the Reactome database (Supplementary Fig. 5d), we found associations to the Ephrin receptor, L1 cell adhesion molecule, and FGF receptor and EGF receptor pathways, each of which have been implicated in GBM invasion[23,24]. From Gene Ontology and Hallmarks, we got Adherens junctions, TGF-beta, and Hedgehog pathways as key hits, all consistent with a migratory and invasive phenotype (Supplementary Fig. 5e–f). We further used transcription factor (TF) target gene sets from CREEDS to identify possible TF regulators of invasion (Fig. 3e). The top hit was *SOX3*, which is consistent with recent evidence[25], and the top 10 list of associations also contained *ASCL1*[26] as well as interesting new suggestions, like *HES6* and *ZEB2*.

Jointly, these findings show that migration phenotypes are strongly associated with known migration pathways. Our data strengthen evidence of a classical subtype GBM cells expressing radial glia markers as particularly invasive and point toward novel TF targets to be explored.

**Most drugs affected glioblastoma cell migration.** Phase 3 of the study also included end-point images of cells treated by 94 different drugs at 11 concentrations. The drugs used in this study were specifically selected in previous phases due to reducing cell viability, as inferred from a metabolic assay. Since accurate diffusion constant estimates are only possible if the net proliferation rate is sufficient, the diffusion constant estimates are unavailable when using cytotoxic drug concentrations. We calculated a threshold for each PDC and drug pair and retained all treated wells with a concentration lower than a calculated threshold. The threshold was calculated to maximize the number of retained wells with high proliferation and removed wells with an insufficient proliferation rate. We only retained PDC and drug pairs with at least three unique treatment concentrations as an additional requirement. Further, drugs with less than 5 retained PDCs were also removed. After filtering, 92 drugs remained.

We estimated the dose-dependent effects of each drug on both the proliferation rate and diffusion constant using a linear mixed-effects model. Such a model is suitable for measuring drug effects on individual PDCs while sharing information across PDCs and drugs. Not surprisingly, all drugs reduced the proliferation rate in a dose-dependent fashion (Fig. 4a). However, more interestingly, most drugs either reduced or increased the diffusion constant, with a clear enrichment of low $p$ values indicating that this was not simply due to random noise (Fig. 4b). After adjustment for multiple corrections, three drugs, dasatinib ($padj < 0.001$), pitavastatin ($padj < 0.01$), and paclitaxel ($padj < 0.05$) significantly reduced the diffusion constant at the $\alpha = 0.05$ level (Fig. 4a). Three more drugs, camptothecin, homoharringtonine, and nilotinib, significantly reduced the diffusion constant at the $alpha = 0.25$ level. Taking dasatinib as an example, we could see a

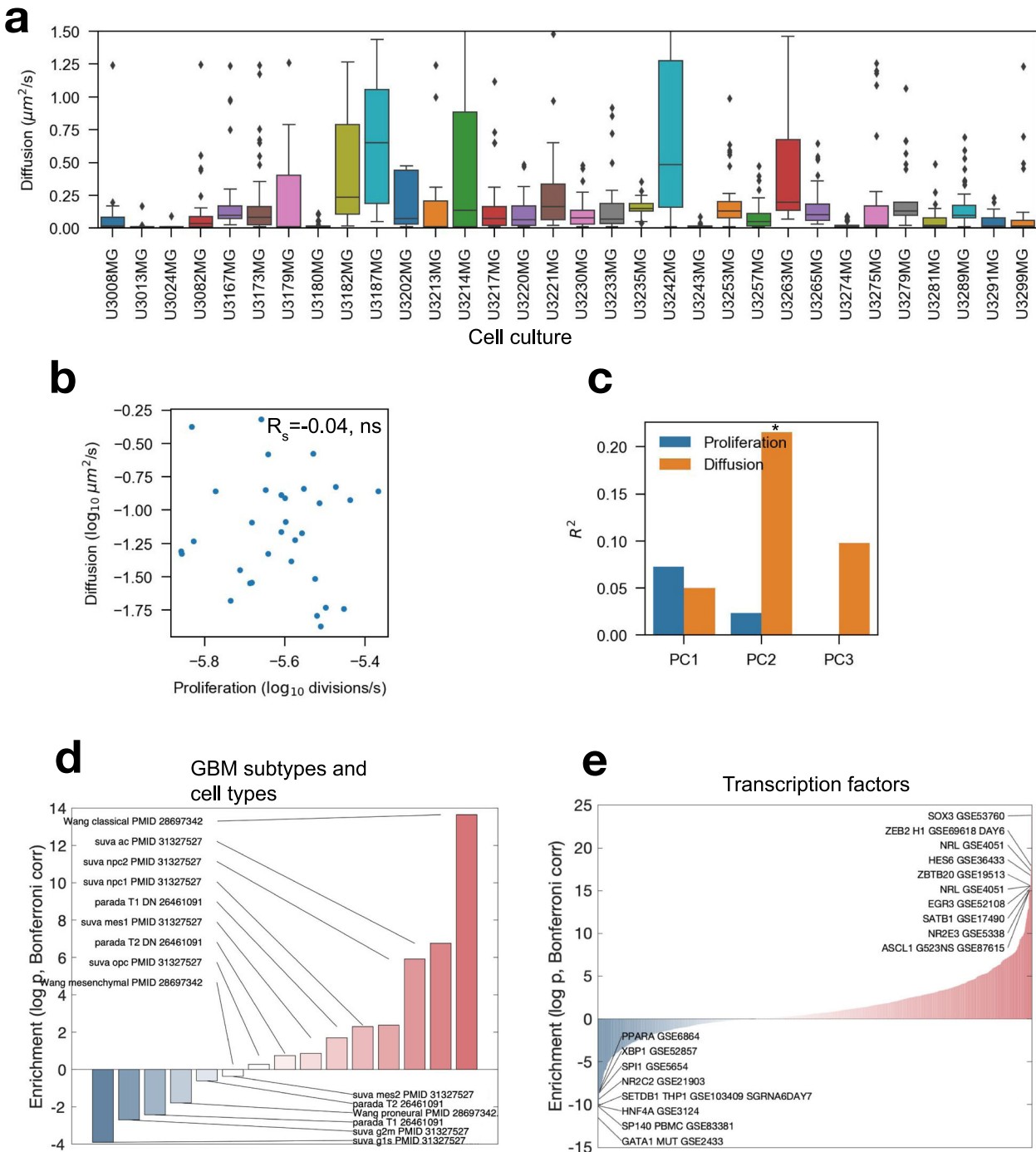

**Fig. 3 Characterization of glioma cell culture specific migration rates. a** The estimated diffusion constant for 32 patient-derived glioblastoma cell cultures (PDCs). **b** The relationship between the diffusion constant and proliferation rate. **c** Spearman correlation coefficient between the first three principal components derived from gene expression data with the proliferation rate and diffusion constant ($n = 32$). **d**, **e** Enrichment scores between the migration signature and (**d**) GBM subtypes and cell types and (**e**) transcription factors. The Y-axis represents signed $log_{10}$ Bonferroni corrected $p$ values. Positive values were positively correlated with the migratory signature.

dose-dependent structural change in wells (Fig. 4c). As the dosage increases, cells become visibly less uniformly distributed. Meanwhile, six drugs significantly increased the diffusion constant at the $\alpha = 0.05$ level. For nine drugs, we also investigated PDC-specific effects on diffusion (Supplementary Fig. 6a), but could not find any meaningful differences.

We found that most drugs affected the PCF and induced a change in cell migration. While all drugs reduced net proliferation, drugs either increased or decreased cell migration.

**Cell tracking confirm anti-migratory drugs.** We used four PDCs to validate the effect of three of the anti-migratory drugs, dasatinib, paclitaxel, and pitavastatin, as well as four of the pro-migratory drugs, colchicine, ciclopirox, nocodazole, and thapsigargin. We failed to acquire one drug, pitavastatin and used simvastatin as a replacement instead. Two GSK3-inhibitors; an indirubin derivative (7BIO), AZD2858, as well as the RhoA inhibitor CCG-1424, previously implicated to have anti-migratory effects in GBM[27–29], were also added to the panel. Cells were imaged every 30 min, tracked,

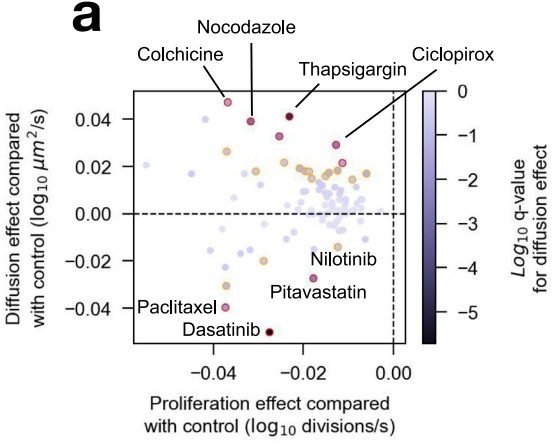

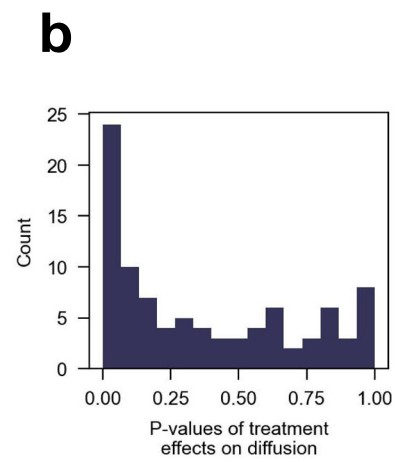

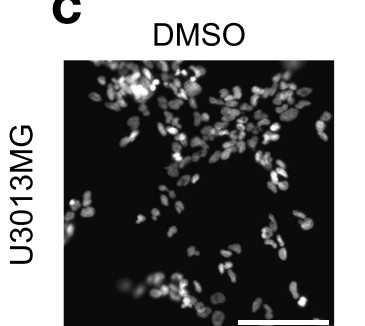
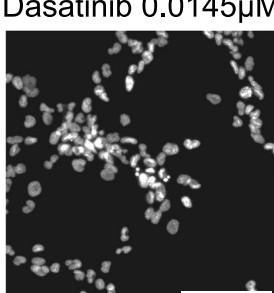
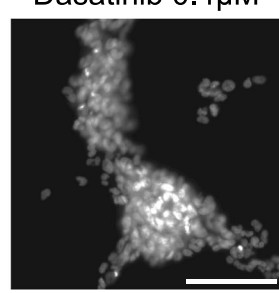
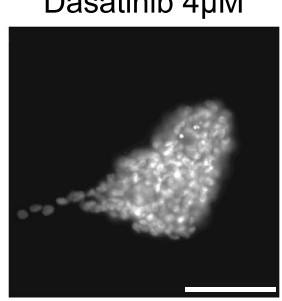

**Fig. 4 Treatments affect both proliferation and migration. a** Treatment effects on diffusion and proliferation for each increment in dose, estimated using a linear mixed-effects model. The color indicates the $log_{10}$ $q$-value of the diffusion effects. Treatments below and above the horizontal dotted line reduce and increase cell diffusion, respectively. Red and orange circled points represent drugs with *padj* < 0.05 and 0.25, respectively. **b** Histogram of *p* values of treatment effects on diffusion. **c** Representative images of dasatinib-treated U3013MG cells. Scale bar is 100 μm. **a, b** $n = 32$ cell cultures and $n = 3$–12 drug concentrations for each treatment estimate (point).

and treated at 11 different concentrations, with two replicates at each concentration. Similar to the untreated validation experiment, we used all images and cell tracking, and not just the end-point, to estimate migration. We first aimed to find concentrations with between 0–20% proliferative capacity compared with untreated controls to identify sub-lethal dosages. We then measured the dose-dependent effect on the diffusion constant at sublethal concentrations (Fig. 5a) as well as cell growth (Supplementary Fig. 6b–k).

Dasatinib significantly reduced migration in three of the four PDCs (Fig. 5b). The exception was U3034MG, which had increased migration at low dosages. Paclitaxel greatly reduced migration at low dosages, but gradually increased migration as the dose increased further (Fig. 5c). We were unable to measure migratory effects at sublethal doses for U3013MG as it was very sensitive to paclitaxel. However, it seemed to respond similarly to the other PDCs. Simvastatin had a strong anti-migratory effect on U3180MG cells ($p < 0.05$, $n = 14$ wells) and a weaker effect ($p < 0.001$, $n = 16$ wells) on U3220MG cells (Fig. 5d). We observed no effect for simvastatin on the other two PDCs. At lethal doses, cells were greatly deformed and nonmotile.

Thapsigargin was predicted to increase migration. However, it reduced migration for three of the PDCs ($p < 0.001$, Fig. 5e). We saw no effect on Nocodazole treated cells (Fig. 5f). Ciclopirox-treated cells showed mixed responses with no clear effect on sublethal doses (Fig. 5g). Colchicine increased the migration for three of the PDCs ($p < 0.05$, Fig. 5h).

Interestingly, all three anti-migratory drugs that were additionally added reduced migration in U3180MG ($p < 0.001$,

Fig. 5i–k). Only AZD2858 reduced migration for multiple PDCs. Comparing the anti-migratory effects with their growth-reducing effects, dasatinib and thapsigargin had the highest magnitude and affected the most PDCs (Fig. 5a).

In the validation experiment, our method identified three out of four anti-migratory drugs. Further, it correctly discarded three drugs that either increased or had no effect on cell migration. The anti-migratory drugs included from previous studies of GBM significantly reduced cell migration in at least one of the four cell cultures. We conclude that our method could significantly improve the selection of drugs chosen for more expensive in vitro or in vivo downstream experiments.

## Discussion

We presented a method for measuring cell growth and migration jointly using end-point images of adherent in vitro cultures. The method applies to images generated from high-content screening experiments and can be retrofitted to pre-existing data sets. It is thus possible to measure cell migration without requiring temporal data or specialized assays, at no additional experimental cost. This was made possible by simulating adherent cell behavior in silico using an agent-based simulation and comparing simulated results with real data using approximate Bayesian computing. Importantly, our method measures migration effects independent of proliferation effects, which is otherwise a common confounder. We applied our method to a large-scale high-content GBM drug screening dataset and estimated both the

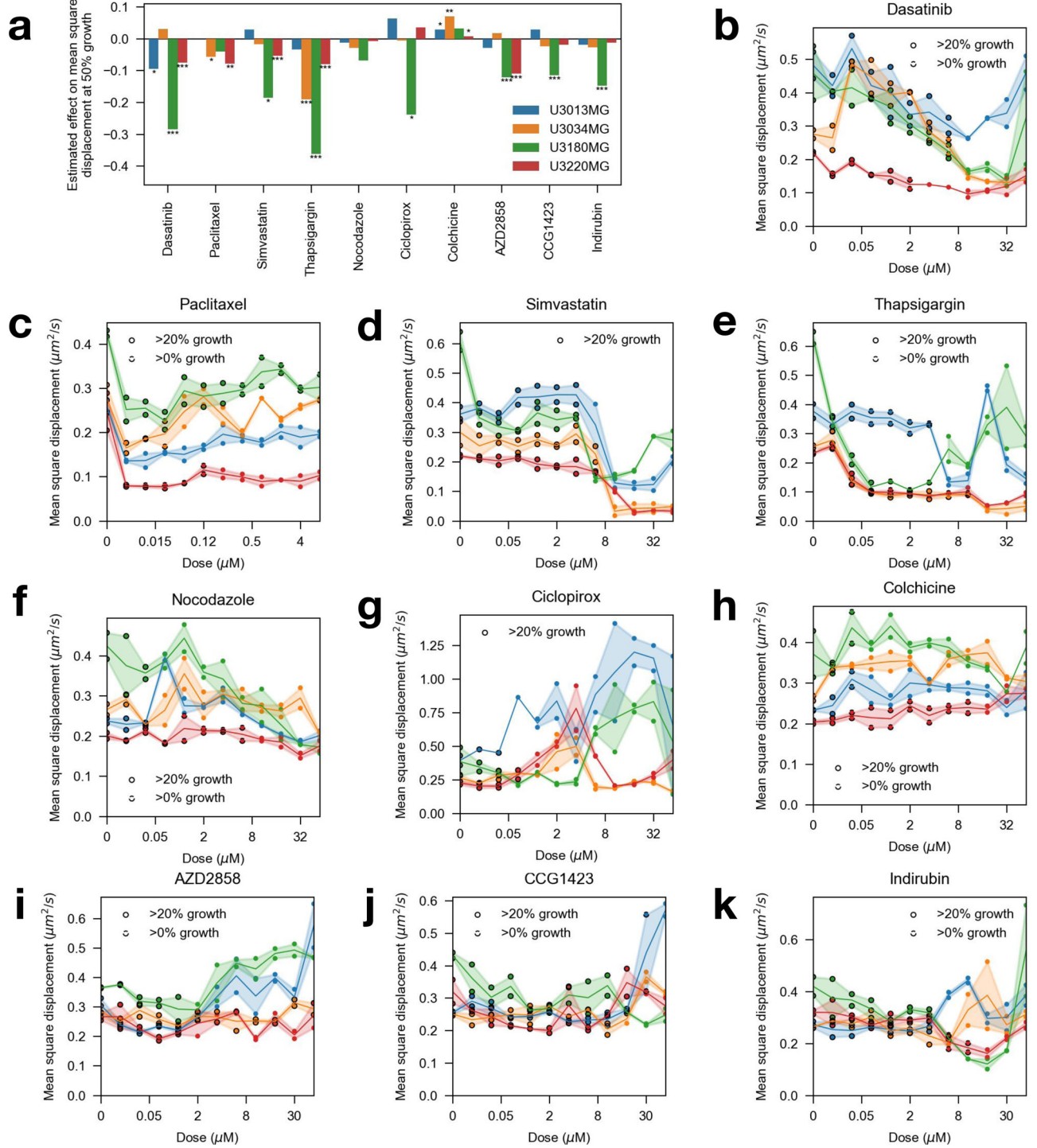

**Fig. 5 Time-lapse imaging confirms anti-migratory treatment effects. a** Treatment effects on cell migration for 50% reduced growth for 10 compounds. Growth refers to the reduced rate of proliferation. *$p < 0.05$, **$p < 0.01$, ***$p < 0.001$. **b–k** Dose-response curves for the diffusion constant for dasatinib, paclitaxel, simvastatin, thapsigargin, nocodazole, ciclopirox, colchicine, AZD2858, CCG1423, and indirubin respectively. Points indicate individual wells while the line shows the mean effect. The colors indicate different PDCs and match the colors and legend in (**a**). Solid and black outlines around points correspond to sublethal doses with less than 20% and 0% growth, respectively. $n = 4$ cell cultures, $n = 24$ wells and $n = 144$ images per well. Shaded regions indicate 95% confidence intervals.

treated and untreated diffusion constant for multiple patient-derived cells and drugs[22]. After filtering, we could estimate the diffusion constant for 32 PDCs and the effect of 92 drugs on cell migration.

Using principal component regression, we found that the second principal component of the gene expression data could

predict migratory behavior. Thus, significant differences in the genome also caused significant differences in adherent migration. Particularly, we find that it is the astrocyte-like or classical subtype that is most correlated with migration, rather than the mesenchymal subtype. Additionally, we identify candidate genes, such as the transcription factor *ZEB2*, that could be driving cell

migration. Although this study was focused on screening for anti-migratory drugs, we believe that these findings warrant further exploration.

We found that most drugs had either positive or negative effects on migration. Thus, it would be beneficial to further explore drugs that both reduce cell growth and migration. By validating our findings using time-lapse imaging, we found that our method could significantly improve our selection of drugs by finding drugs with multiple beneficial effects. Our method identified dasatinib as a promising drug, reducing both cell growth and migration. Dasatinib is known for its anti-migratory effects in multiple cancers, including GBM[30,31]. It has previously been investigated in a phase 2 trial aimed at reducing invasion induced by the anti-VEGF drug bevacizumab, although it failed to improve survival[31]. We found that another tyrosine kinase inhibitor, nilotinib, also reduced migration. Interestingly, we found that paclitaxel, which hyper-stabilizes microtubules, also reduced migration. Contrary, drugs which depolymerize microtubules, such as colchicine and nocodazole, instead increased migration. Paclitaxel is known to reduce migration in GBM but has been of limited application so far as it does not cross the blood-brain barrier[32,33]. We additionally found that statins show an anti-migratory effect and could be explored further.

Our method is based on two requirements to work. First, cells need to proliferate to measure diffusion. Slowly growing cells do not form large enough clusters to detect changes in the pair correlation function. In our study, we found that 78% had high enough proliferation. Additionally, 55% of the PDCs with low proliferation had practically no growth, indicating that these cells should not be considered a good model for GBM regardless of the assay. Hence, we were able to measure the diffusion of 90% of the relevant PDCs included in the study. However, this requirement also prevents the method from estimating cell migration at lethal drug concentrations. We argue that anti-migratory effects matter the most at sublethal concentrations. At lethal concentrations, the cells are effectively neutralized or dying, and reducing migration will likely be of low benefit. Second, the model assumes Brownian motion. However, cells might cluster not because of proliferation, but also due to cell-cell adhesion or attraction. From studying the cell movement in the time-lapse experiments, we found no relevant cell-cell adhesion or attraction in nine PDCs (Supplementary Fig. 3c, d). But, in one PDC, we observed excessive clustering due to cell-cell interactions Supplementary Fig. 3e. In this case, the image analysis failed to recognize single cells and we were unable to measure either migration or proliferation. In the drug-screen experiment, we automatically identified images where single cells could not be identified, and removed them from the study. Cells may also move persistently in one direction before changing direction. However, we found that this was not relevant for time spans longer than a couple of hours in our case. Conclusively, we found that our method is applicable to the cells and data set used in this study.

A challenge with studying migration in vitro is that 2D adherent cancer migration is not representative of invasion in the brain. In patients' tumors, cells move in 3D and have to break down or circumvent the extracellular matrix to progress. In particular, for GBM, invasion is challenging due to narrow routes in the brain parenchyma and requires cells to perform drastic cell size changes to invade[34]. Additionally, cell motion is affected by local anatomy and environmental variables. For example, cells might be attracted to and co-opt local vasculature[35,36] while migrating away from hypoxic regions[37,38]. Neither of these factors is present in adherent cultures. Nevertheless, our migration estimates correlate with migratory signatures and drugs associated with in vivo invasion. For example, our top hit, dasatinib, has been shown to reduce GBM invasion in orthotopic

xenografts[39]. One recent study also suggests dasatinib as a precision treatment for GBM patients with mesenchymal subtype and high Src activation[40]. We found that GBMs belonging to the classical subtype were associated with invasion. This is consistent with recent findings by our team, in which classical GBMs invaded diffusively in a zebrafish model[41].

The existence of resistant subpopulations within a single well is not accounted for in our method and could potentially affect its estimates of proliferation and migration rates. The impact of resistant subpopulations could be investigated in the case of time-lapse data by estimating the proliferation and migration rates individually for each cell. In addition, one could trace the lineage of the cells, which could reveal resistance in terms of elevated proliferation rates in certain sub-lineages. We reserve such extensions for future work. We can also expect the population of GBM cells to exhibit cell-to-cell variation in migration speed. This could, in part, explain the outliers in Fig. 2e, where some estimates of diffusion coefficients differ by an order of magnitude between the two methods.

One advantage of our method is that it allows for measurements of drug effects on migration in a high-throughput setting, in which a high number of drugs are covered over multiple doses and cell lines. Our data strongly suggests patient differences in how migration is affected by drugs. For instance, our method could detect differences among previously identified anti-migratory drugs across patient-derived cell lines. These compounds included two GSK3 inhibitors (indirubin and AZD2858) and a RhoA inhibitor (CCG-1423). Previous data for these compounds, collected across a limited number of doses and GBM cell lines indicate effects on cell migration[27–29]. Of note, CCG-1423 can induce a mesenchymal-amoeboid switch in 3D cultures[27]. Further work will be needed to fully explain the differences between individual patient-derived cell cultures, and evaluate effects in 3D assays, reserved for future work.

There are many foreseeable extensions of the method. For instance, the addition of images of the initial distribution of the cells has the potential to improve the accuracy of our method. With such data, it would be possible to estimate the initial pair correlation function (PCF), and this can be used to inform the initial spatial distribution of cells in the agent-based model. Thus, instead of assuming a spatially homogeneous distribution one would place the cells such that the PCF is as similar as possible to the initial PCF for each well. This procedure eliminates the assumption of spatial homogeneity and should improve the estimates obtained from the model. The ABC method can also likely be extended. In this investigation, we opted for a variant of ABC where the top 0.5% samples from the prior are retained to form the posterior distribution. This was due to the large variation in the discrepancy in summary statistics between simulations and the experimental data. However, this choice made it more difficult to draw conclusions about the goodness of fit, and in future work, it might be worth considering more recent versions of rejection sampling that make use of adaptive tolerance, see e.g., ref. [42].

The simulation and estimation package used in this article is provided as a ready-to-use Python package (github.com/emilrosen/endpoint_cell_diffusion). The method should be used in the early stages before findings are validated in more expensive in vitro and in vivo models. We expect our method to extend the functionality of high-content screens to also screen for anti-migratory drugs.

## Methods

**Human glioblastoma cell cultures**. Human glioblastoma multiforme patient-derived cells (Supplementary Data) were obtained from the Human Glioblastoma Cell Culture (HGCC)[20] biobank and maintained in laminin-coated T25 flasks

(Corning catalog #353808) in a 1:1 nutrient mix of Neurobasal and DMEM/F12 (Gibco catalog #21103-049, #31331-028) medium supplemented with B27 and N2 (Thermo Fisher Scientific catalog #12587-001, #17502-001) and human recombinant EGF and FGF (10 ng/ml, Peprotech). The cell cultures were between passages 8 and 24 (average 16), see also ref. [22]. TrypLE select (Gibco catalog #12563-011) was used to dissociate cells before seeding adherent cultures into plates for experiments. Note that cells must be fully dissociated before seeding for the assumptions of our algorithm to hold. The number of cells seeded was optimized to achieve a subconfluent growth phase (approximately 70 percent) at the end of the assay. All cell lines were checked by Mycoalert kit (Lonza catalog #LT07-418) and displayed no mycoplasma contamination.

**Data acquisition of time-lapse validation experiment of untreated cells**. 10 HGCC cell cultures (Supplementary Data) seeded in five 96-well Primaria plates (VWR 734-0079), with two PDC per plate. Each PDC was seeded at 2000, 1000, 500, 250, and 125 cells per well with 8 replicates each. Plates were put in the Incucyte S3 (37 °C, 5% CO$_2$) and imaged at 10x magnification once every hour over 4 days.

**Data acquisition of time-lapse validation experiment of treated cells**. Four HGCC cell cultures (Supplementary Data) were seeded in two replicates in 96 well primaria plates (VWR catalog #734-0079) coated with laminin at a density of 2500 cells/well and grown in a 5% CO$_2$ incubator at 37 °C. Dasatinib, Simvastatin, Ciclopirox ethanolamine (Selleckchem catalog #S1021, #S1792, #S3019), Thapsigargin (Sigmaaldrich catalog #T9033), Nocodazole, Colchicine, (MedChem Express catalog #HY-13520, #HY-16569) were reconstituted with DMSO at a stock concentration of 10 mM. After 24 h of cells seeding, above drug treatments were added in the concentration range of 0.005, 0.05, 0.5, 1, 2, 4, 8, 16, 25, 32, and 50 µM, Paclitaxel (MedChem Express catalog #HY-B0015 reconstituted in DMSO, 10 mM) in the concentration range of 0.007, 0.015, 0.031, 0.06, 0.125, 0.25, 0.5, 1, 2, 4, and 8 µM, AZD2858, Indirubin (7BIO), and CCG-1423 (MedChem Express catalog #HY-15761, #HY-121035, #HY-13991, reconstituted in DMSO, 10 mM) in the concentration range of 0.005, 0.05, 0.5, 1, 2, 4, 8, 16, 25, 30, and 40 µM. Plates were transferred into the IncuCyte S3 soon after adding the treatments for live image acquisition (10X magnification) every 30 min for up to 72 h.

**Data acquisition of drug screen images**. Data were acquired from ref. [22] using end-point images from phase 3. In brief, 41 patient-derived cell cultures from the Human Glioma Cell Culture resource[20] were suspended in stem cell medium and plated on 384 well plates (BD Falcon Optilux #353962) coated in laminin at a density of 1000–2200 cells/well. The cells were cultured at 37 °C, and 5% CO$_2$ for 96 h and imaged using the ImageXpress microscope at 20x magnification. Treatments were added after 24 h in treated wells using 11 doses and 94 drugs. Images consisted of two fluorescent channels (cell nuclei and cytoplasm), and a phase-contrast channel (see ref. [22] for details). During the imaging process, the plates were washed, which also removed dead cells. It was, therefore, not possible to reliably identify dead cells in the images.

**Image segmentation**. Two deep convolutional neural networks (DCNN), based on the U-Net architecture[43], were used for cell segmentation for the images from the time series validation experiments and drug screen images. The two DCNNs were trained on different training data due to different cameras, instruments, and channels used, but were otherwise identical (Supplementary Data). We implemented the DCNN in Python 3.5.3 using Keras 2.1.6[44] with Tensorflow 1.8.0[45] as the backend. The DCNN was trained to separate distinct cell bodies by weighting pixels between two adjacent cell bodies higher. Additionally, the DCNN applied to the drug screen images was also trained to identify areas with non-adherent growth. After segmentation, we used the watershed algorithm to find cell positions. The total number of cells was counted and the PCF calculated for each well (Supplementary Data).

**Cell radius estimation**. Cell radius was estimated from the images using the segmented images (Supplementary Fig. 2). The area for each cell was calculated, followed by calculation of the radius calculated by treating the cell as a disk. Note that these cell radius estimates are not needed for the model and were only used to compare the results with the cell radius estimates using the ABC method.

**Analysis of time series data**. Cell bodies were linked between time points for the time series images using the python library Trackpy 0.5 and python 3.6.2[46,47]. Using trackpy, we computed the MSD for each well, using a max lag time of 24h. The diffusion coefficients were estimated by $MSD = D \cdot t$, where $D$ is the diffusion coefficient, and $t$ is the lag time[21]. Point estimates of the PDC-specific diffusion constants were calculated by the mean value of the diffusion constant of each well.

**Individual-based model**. We model the well in which the cells migrate, divide, and die as a square region in two-dimensional space with linear size $L$, denoted

$\Omega = [0, L] \times [0, L]$. The position of cell $i$ is denoted $x_i(t)$ and to denote the position of all cells we write $\mathbf{x}(t) = \{x_i(t)\}_{i=1}^{N(t)}$, where $N(t)$ is the number of cells at time $t$.

We assume that cell motion is over-damped and therefore model cell migration and mechanical interactions as a Langevin equation[48]:

$$dx_i(t) = F_i(\mathbf{x}(t))dt + \sqrt{2D}dW(t), \tag{1}$$

where migration is modeled as a Brownian motion ($dW(t)$) with diffusion coefficient $D$ and intercellular forces (pushing/adhesion) are captured by the drift term $F_i(\mathbf{x}(t))$, which depends on the position of all other cells. The forces between cells are assumed to be pairwise, and the total force experienced by cell $i$ can hence be written as

$$F_i(\mathbf{x}(t)) = \sum_{j \neq i}^{N} f(|x_i - x_j|) + f_{ext}(x_i), \tag{2}$$

where $f(r) = f_0$ if $r < R$ (the cell radius) and zero otherwise. The term $f_{ext}(\cdot)$ represents external forces imposed by the wall of the well.

We assume that cell division occurs with a PDC-specific base rate $\alpha > 0$, and is reduced due to contact inhibition by the presence of other cells. This is modeled using an interaction kernel $w_b(r) = e^{-\gamma r}$, which only depends on the distance $r$ between the cells. The total rate of cell division for cell $i$ with position $x_i$ is given by

$$\rho_i = \alpha - \beta \sum_{j=1, j \neq i}^{N} w_b(|x_i - x_j|) \tag{3}$$

Upon cell division, the daughter cell is placed at distance $R$ from the parent cell, and at an angle drawn uniformly from the interval $[0, 2\pi]$. In this study, we set $gamma = 1$ and $beta = 0.8$.

Lastly, we assume that cell death occurs at a rate $\mu$ independent of cell density. We set $\mu = 0$ for the drug screen and time-lapse validation experiments.

**Numerical implementation of simulation**. Each image represents a cropped part of each well close to the center of the well. To reduce computational time, we used an artificial well with a wall length of 1.3 times larger than the camera field of view. Thus we could reduce the number of cells in the simulation while keeping density equivalent, assuming that any potential edge effects of the reduced well were negligible. Other parameters were kept as close as possible to the physical properties used in the experiment. See Supplementary Data for exact parameter values.

The system is initialized by placing $N_0$ cells at random in the domain, with coordinates drawn from a uniform probability distribution on the interval $[0, L_s]$ for both the $x$- and $y$-coordinates. The fact that the model contains both discrete and continuous dynamics is handled by coupling the Gillespie algorithm[49], which is used for handling discrete birth/death events, to the numerical solution of SDEs (using the Euler–Maruyama method[50]) that describe the motion of the cells. The simulations were carried out for 96 h using a time-step of 50 s. We used KD-trees[51] to find nearby cells and speed up cell-cell interactions, which would otherwise scale with $\mathcal{O}(n^2)$. Simulations with a high proliferation rate would quickly reach confluence and significantly increase simulation length. We discarded simulations with an unrealistic number of cells (2 times max number of cells in the most populous well) and let those simulations stop early. At the end of each simulation, we applied a virtual camera mimicking the field of view of the real camera and only kept cells within these regions. The number of cells and the PCF were calculated in both of these regions. The simulation was implemented in Python 3.6.2 and is made available as a GitHub repository (github.com/emilrosen/endpoint_cell_diffusion).

**Parameter estimation**. To find the model parameters that best describe the data, we used Approximate Bayesian computation (ABC)[19]. Two sets of simulations were created, each with parameters matching the experimental parameters of the time series validation experiment and the drug screen data. Both sets of data contained wells with a different number of initially seeded cells. For both the time-series validation experiment and the drug screen experiment, we generated 100,000 simulations for each unique seeding density. Due to the large number of wells, we reuse simulations for wells with identical initial conditions.

For each simulation, we varied three parameters; proliferation rate, diffusion constant, and cell radius (Supplementary Data). As summary statistics, we used the final cell count and the PCF. The error for cell count and PCF, respectively, were defined as

$$E_n = \left( \frac{N_w - N_s}{1 + N_w} \right)^2 \tag{4}$$

$$E_{pcf} = \frac{1}{K} \sum_{i=1}^{K} \left( \frac{PCF_w(i) - PCF_s(i)}{1 + PCF_w(i)} \right)^2 \tag{5}$$

for well $w$ and simulation $s$ where $K$ is the number of bins used to approximate the PCF. The final distance function was defined as $E = \sqrt{E_n + E_{pcf}}$.

**Convergence analysis**. Typically in ABC estimation, all simulations with $E < \epsilon$ are retained to form the posterior. However, we found that setting a singular threshold for all samples and wells did not work well in practice. In addition, from the

simulated data, we noted that the errors from the ABC method ($E$), were only marginally related to the error between the known and estimated parameter values. A higher $E$, indicated slightly higher parameter errors as well, but a majority of the variation of $E$ was still unexplained (linear regression $R^2 < 0.2$, Supplementary Fig. 7A–C). Instead, we opted to retain 0.5% (500 simulations) of all simulations with the lowest error. We found that this method worked better in practice, as long as our bank of simulations was large enough.

For the simulated data, to determine if the approximated posteriors were suitable, we first investigated how the parameter errors changed by varying the acceptance ratio. Decreasing the acceptance ratio also gradually reduced the parameter errors (Supplementary Fig. 7D–F), as the peak of the posterior distributions better reflected the true values used. There were very small gains by reducing the acceptance threshold below 1%. Further, we investigated how convergence of the diffusion constant would be affected if we introduced a non-zero death rate into the simulations since the diffusion estimates were negatively affected by reduced proliferation. Similar to (Supplementary Fig. 7E), the errors decreased rapidly and stabilized well before the acceptance threshold used in the article (Supplementary Fig. 7G). However, the error was also consistently higher, indicating that the death rate made it more difficult to get correct estimates, regardless of how many simulations were used. We omitted proliferation rate as we already noted that estimates would reflect net proliferation, but otherwise be accurate, when introducing cell death.

For the drug screen experiment, the true parameter values were unknown. Instead, we sought to look at how stable the parameter estimates were while varying the acceptance ratio. If the parameter estimates change drastically for different acceptance thresholds, then the posterior would not be stable and we would need to increase the number of simulations used. We used the parameter estimates using the 0.5% acceptance ratio as a reference and compared how different the estimates would be if we instead had chosen another acceptance ratio (Supplementary Fig. 7H, I). The estimates were stable with only minimal changes to the parameter estimates below 10%. Our proposed simulation bank was thus large enough to form consistent parameter estimates.

**Point estimates of model parameters**. The ABC method generates an approximated posterior for each model parameter for each well. Due to the uniform prior used in the model, we opted to use the mode rather than the mean value for point estimates. Each parameter posterior was divided into 11 equidistant bins, and the mode was assigned as the bin containing the highest number of retained parameter values. PDC point estimates were calculated by taking the mean across all well-specific point estimates.

**Differentially expressed genes and gene sets**. The HGCC contains RUV-normalized Affymetrix gene expression data for 100 PDCs[22], including the 41 PDCs in this study. The $p$-value histograms were calculated using a linear regression between gene expression and the estimated log proliferation and diffusion estimates ($n = 32$ PDCs, $n = 23832$ genes). Three principal components (PCs) were derived using all 100 PDCs in the data set and principal component analysis to reduce the dimensionality. The correlation between each PC and the log proliferation and diffusion estimates were computed ($n = 32$ PDCs).

Single sample gene set enrichment analysis[52] was used to get a gene set enrichment score for each PDC and gene set. The Spearman correlation coefficient was calculated between the second PC and each gene set/gene of interest ($n = 100$ PDCs). Adjusted $p$ values ($q$-values) were computed using the Benjamini-Hochberg correction. Adjusted $p < 0.01$ were considered significant.

**Estimation of drug effects**. We estimated drug effects on the model parameters using the linear mixed-effects model

$$Y \sim 1 + Drug : Concentration + (1 + Drug : Concentration || PDC)$$

where Y is either the logged proliferation rate of the diffusion constant, Drug is the drug used, Concentration is the logged treatment concentration used, and PDC is the cell culture. Thus, the model estimates the fixed linear effect on parameter Y for each drug across all PDCs, while sharing intercept values for each PDC across all drugs. PDC-specific responses are calculated as random effects. A drug was considered a statistically significant effect on parameter Y if the Benjamini-Hochberg adjusted $p < 0.25$. The model was implemented using statsmodels 0.12, and python 3.6.2[53].

**Statistics and reproducibility**. The number of data points underlying each analysis is given separately, above. All cell culture experiments were carried out in 2–8 replicates, as described above. The Bonferroni method was used to correct $p$ values when testing for gene set enrichments.

**Reporting summary**. Further information on research design is available in the Nature Portfolio Reporting Summary linked to this article.

## Data availability
The imaging data is available from the authors upon request. The simulation was implemented in Python 3.6.2 and is made available as a GitHub repository (github.com/emilrosen/endpoint_cell_diffusion).

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

## Acknowledgements

We thank the Human Glioma Cell Culture consortium (hgcc.se) for the development of the cell cultures. Figure 1a, b was created with BioRender. This work was supported by the Swedish Cancer Society (200839PjF), the Swedish Research Council (20180269), and the Swedish Foundation for Strategic Research (BD15-0082).

## Author contributions

P.G. and S.N. conceived the study. H.M., L.E., R.S., and C.K. performed the validation experiments. E.R. performed all analyses and wrote the first draft of the manuscript. All authors contributed to the final version of the manuscript. P.G., C.K., and S.N. supervised the project.

## Funding

## Competing interests

The authors declare no competing interests.
