## [Peer Review File · Communications Biology]

Reviewers' comments:

Reviewer #1 (Remarks to the Author):

Summary

This is a really nice paper based on a clever idea that the spatial positioning of cells (in culture) is a readout of the cell's migratory behaviour. The idea is that proliferating cells that have high migration rates are more dispersed than their slow-moving counterparts – all else being equal. The authors construct a simple agent based model to simulate the spatial patterning of cells expected given a particular birth, death and migration rate. The Pair Correlation function (PCF) is used to quantify the spatial dispersion. They show using simulated data (where parameters are known) that Bayesian inference (ABC) can reasonably well recover the correct model parameters, and then apply their methodology to infer glioblastoma cell line migration rates from images of wells collected during a high throughput screen. Inferred migratory behaviours correlate with gene expression signatures of migratory cells. Finally the authors experimentally verify their inferred migration rates using live imaging.

Overall it is a very thorough piece of work with good testing and validation of the modelling results. We especially enjoyed the simplicity and elegance of the approach, which is nevertheless really quite powerful. As the authors explain, this methodology could be relatively easily rolled out to reanalyse high-throughput drug screening in a very broad variety of settings (and institutions).

We look forward to seeing it in print.

The comments below are all quite technical and we hope they might be useful to improve what is already a very high-quality piece of work.

1. In the ABC, the authors do not use a rejection step for any simulation where the distance statistic falls below some tolerance threshold. Instead, they retain the 1000 closest simulations to the data. Because this approach does not penalise parameter combinations that lack confidence in the posterior estimates, more attention should be given to some spread statistic of the posterior distributions. For example, an analogous plot to Fig 2B could instead show the real vs estimated values, but instead show points shaded by the variance of the posterior distribution to illustrate where the model loses confidence. It would also be good to plot quantiles of the posterior distribution against the acceptance threshold and show that the acceptance threshold is sufficiently small for the posterior to have converged.
2. Related to the previous point, with additional simulations the authors show that allowing a positive death rate ($\mu > 0.0$) does not bias the estimate of the diffusion parameter (Supp Fig 1G). It would also be informative to know whether including a death rate decreases the accuracy of the inferred diffusion rate by plotting the death rate vs the quantiles of the posterior distributions (and ground truth).
3. In 'parameter estimation' (in methods): '...we reuse simulations for wells with identical conditions'. Given our comment (1) the authors should show that their bank of simulations is large enough for this to be OK (showing posterior convergence would probably suffice).
4. When using their approach to investigate the effects of compounds on cellular migration, there is no discussion on the impact of potential resistance phenotypes. The mode of resistance inheritance could dictate the spatial distribution of cells that remain: for example, if cells rapidly transition in and out of a resistance via a non-genetic mechanism, one might expect resistance to be more spatially diffuse, spread more evenly amongst dividing cells. This could impact the expected spatial distribution of cells post-treatment and hence has the potential to bias the model's diffusion estimates. Whilst the authors choose 'non-cytotoxic' concentrations of compounds, this is only achieved via choosing wells with a sufficiently high proliferation rate – however this is the net rate for the entire well. It is possible that this net proliferation is achieved by a sub-population of resistant cells. Could these issues either be addressed in the model or excluded via analysis of the time-series data during drug exposure?
5. How reliable is the cell segmentation DCNN? If certain compounds influence the morphology of cells, could the networks mistakenly call larger dysmorphological cells as a group of smaller cells? This would interfere with the model inference. Some example data of treated and non-treated cell segmentations would help convince readers that this approach can confidently identify individual cells.

6. How is the radius of a cell calculated, given cells are often not spherical? As the PCF works by drawing a circle of radius r around the centre of the cell, cell morphology could bias the statistic and, by turn, the model. Related to the previous point, could some example estimated radii be shown overlaid on some incuocyte images?
7. We don't think there is a mention of the necessity of deriving a single cell solution at the beginning of the experiment. Presumably this is important, as without it the in vitro experiment would break the model assumption that cells are distributed uniformly throughout the well at $t=0$. Whilst this might seem obvious, deriving single cell solutions can sometimes be challenging; a brief point highlighting this would aid the adoption of this approach by other experimentalists.
8. As a suggestion for future work, not necessary for this paper, (related to the above) is the method more reliable if before and after images are used (rather than just after images). We can imagine it could help particularly in the setting when proliferation rates are relatively low.
9. In Fig 2, the description of A and B in the legend appear to be the wrong way around.
10. Some of the units are written incorrectly in the axes labels – for example, in Fig 2, diffusion is given as μ^2 / s – I think this should be $\mu m^2 / s$ (micrometres² per second? Opposed to μ which denotes the death rate in the model).
11. In Fig 2E there are estimates from the time-series data that are orders of magnitude higher than those from the end-point alone images. As the time-series estimates are presumably more accurate, we think these outliers deserve comment.
12. An identity line would be helpful on plots that are comparing true and estimated parameters where the axes limits differ (Fig 2B, for example).

Review by Trevor Graham & Freddie Whiting

Reviewer #2 (Remarks to the Author):

In this manuscript, the authors describe a method to estimate cell viability and migration from single end-point images of patient-derived glioblastoma cells and propose it as a method for standard drug screening. The proposed method offers the possibility to measure cell migration without requiring temporal data (such as, time lapses) or specialised assays (including scratch assay or Transwells-based migration assays). The authors offer the method as an open software. Overall, the manuscript describes an interesting method with a potential application in the study of cell migration, which could be potentially used in standard drug screening.

A few general questions arise after reviewing the manuscript.

1. The authors claimed screening of cell viability, however there are no measurements of viability in this manuscript. Cell death is mentioned, but the only outcomes presented are proliferation and diffusion rates. In fact, it is stated in a part of the study that death rate was not able to be estimated. How does this affect the robustness of the method? Therefore, claims about cell viability should be toned down and implications of not accounting for cell death should be clearly stated.
2. Additionally, the method seems to be limited when cells have a low rate of proliferation. Thus, when screening potent drugs that affect cell proliferation pathways, it may not be possible to use this method to assess these drugs limiting the application to drug screening.
3. In drug screening, when screening a drug in a concentration-response mode, certain parameters, such as IC50 values, are key outcomes. Concentration response curves were plotted in Figure 5 from the data obtained from a time-lapse experiment. However, no data was shown for the end-point analysis using this new method. Measurement outcomes described along the manuscript are proliferation and diffusion rates. How relevant are these in the context of drug screening? Would be relevant to calculate and compare IC50/EC50 values using the end point analysis and time lapse.

4. In the initial part of the manuscript (Figure 1), when validating the method, the authors compared the raw image and the simulation images. In cells with a low diffusion, these two images do not show a clear overlapping with a discrepancy at shorter distances. Based on this, it seems that for cells with a low diffusion, the correlation is not as accurate. Does this affect the overall analysis of proliferation and diffusion? Is the method reliable for such cultures?

Specific minor points that might need to be addressed:

1. Line 76: the results start by addressing Figure 1C with no mention to Figure 1A-B till later in the results section. The images are showing up in order.

2. Line 79: Spatial distribution may vary between wells. The same cell line should show similar distribution across the wells unless the seeding was not properly done. One would expect the same distribution across different wells.

3. In general, figure legends are very brief and lack specific details. For example, in Figure 1, low and high diffusion cells correspond to which cell line? No details about the name of the cell line, passage, and other details. Also, no scale bars on any of the images. Same for Figure 4C.

4. Line 90: It is stated in the text that cells with low cell diffusion tend to cluster together, while high diffusion do not form clusters. There is no reference that supports this statement.

5. Figure 5, concentration is represented as $\mu\text{M}/\text{L}$, which is incorrect. Does it mean concentration in μM ? Graphs are not very intuitive. Numbers in the bar graphs correspond to the cell ID?

6. Overall, when referring to supplementary data there is no specific indication of the Figure that the text is referring to. If data in supplementary information is cited in the main text, it should be clear to which particular dataset the author is referring.

7. The correlation between the transcriptomic profile of the glioblastoma cells and migration response are interesting, however the authors offer a very superficial analysis with no in-depth analysis. Even though the aim of the manuscript it to describe the new analysis method, if the data is introduced, it should present detailed analysis.

Reviewer #3 (Remarks to the Author):

The research by Rosen et al describes the development and initial validation of a methodology to estimate cell motility from single end-point images in vitro using a range of GBM cell lines and a panel of inhibitors. I enjoyed reading this manuscript and would like to see it published but have a few comments I would like to see addressed:

1. What is the rationale of the choice of inhibitors? There is a mention of 'promising' drugs. Are these clinically relevant drugs as it seems? It would be very interesting to include anti-migratory drugs identified and included in recent GBM studies, such as indirubin derivatives BIO-indirubin or AZD2858; or newly developed drugs such as CCG-1423, alongside the commercial drugs used that had been developed as cytotoxic or anti-proliferative (eg Dasatinib) drugs. It would be interesting to see if there are differences in the effect of the drugs on cell migration using pre-defined anti-migratory drugs.

2. Can you comment on observed phenotypes and the effect on cell migration? In figure 4C it looks like the cells are becoming more adherent and sheet-like with increasing drug concentration and may display a switch to a collective type of cell migration? With changes in their migratory behaviour?

3. Can you also comment on potential-mesenchymal to amoeboid transition in response to drug treatment and knock on effect on cell migration; this would explain some effects on cell migration opposite to what was expected?

4. Can you summarise the findings for the cell lines you used in terms of the subtypes, as these there representatives of all three subtypes and can this information be included in a table?

5. The paragraph in the discussion relating to 3D assays needs expanding (page 10, line 371 onwards). Data generated in 2D may not be representative of data generated in 3D. Have you started to look at analysing data generated, for example, in 3D invasion assays (is this possible in your set-up?)? I was thinking particularly with regards to the results from the migration signatures and the transcription factors. Is there more evidence from the literature to support the biological and clinical relevance of these findings?

6. When will this application made freely available?

7. Minor comments:

In the result section (first result); I think the mentioning of the figures in text should be in order, so Figure 1A should be mentioned first then 1B etc.

Can you check gene names are in italics?

Point-by-point response to reviewers' comments:

Reviewer #1 (Remarks to the Author):

This is a really nice paper based on a clever idea that the spatial positioning of cells (in culture) is a readout of the cell's migratory behaviour. The idea is that proliferating cells that have high migration rates are more dispersed than their slow-moving counterparts – all else being equal. The authors construct a simple agent based model to simulate the spatial patterning of cells expected given a particular birth, death and migration rate. The Pair Correlation function (PCF) is used to quantify the spatial dispersion. They show using simulated data (where parameters are known) that Bayesian inference (ABC) can reasonably well recover the correct model parameters, and then apply their methodology to infer glioblastoma cell line migration rates from images of wells collected during a high throughput screen. Inferred migratory behaviours correlate with gene expression signatures of migratory cells. Finally the authors experimentally verifying their inferred migration rates using live imaging.

Overall it is a very thorough piece of work with good testing and validation of the modelling results. We especially enjoyed the simplicity and elegance of the approach, which is nevertheless really quite powerful. As the authors explain, this methodology could be relatively easily rolled out to reanalyse high-throughput drug screening in a very broad variety of settings (and institutions).

We look forward to seeing it in print.

The comments below are all quite technical and we hope they might be useful to improve what is already a very high-quality piece of work.

Response: Thank you for the kind words. We would like to thank the reviewer for the valuable comments, which have helped us in improving the manuscript.

- 1. In the ABC, the authors do not use a rejection step for any simulation where the distance statistic falls below some tolerance threshold. Instead, they retain the 1000 closest simulations to the data. Because this approach does not penalise parameter combinations that lack confidence in the posterior estimates, more attention should be given to some spread statistic of the posterior distributions. For example, an analogous plot to Fig 2B could instead show the real vs estimated values, but instead show points shaded by the variance of the posterior distribution to illustrate where the model loses confidence. It would also be good to plot quantiles of the posterior distribution against the acceptance threshold and show that the acceptance threshold is sufficiently small for the posterior to have converged.*

Response. Thank you for the comment. We certainly agree that it would be beneficial to also study the spread statistics and not only the point estimates of the posterior. Still, applying our method in practice, the uniform prior means that the posterior is zero outside of the interval. This, in turn, means that variance and quantiles of estimates close to the boundaries are affected in a way that makes it hard compare the spread statistics between

samples. Responding to this and the below points, however, we have now added **Supplementary Figure 7** to document robust convergence of the posterior, and also added a section, 'Convergence analysis' to Methods, (**page 16**):

*"Typically in ABC estimation, all simulations with $E < \epsilon$ are retained to form the posterior. However, we found that setting a singular threshold for all samples and wells did not work well in practice. In addition, from the simulated data, we noted that the errors from the ABC method (E), were only marginally related to the error between the known and estimated parameter values. A higher E , indicated slightly higher parameter errors as well, but a majority of the variation of E was still unexplained (linear regression $R^2 < 0.2$, **Supplemental Figure 7A-C**). Instead, we opted to retain 0.5% (500 simulations) of all simulations with the lowest error. We found that this method worked better in practice, as long as our bank of simulations was large enough.*

*For the simulated data, to determine if the approximated posteriors were suitable, we first investigated how the parameter errors changed by varying the acceptance ratio. Decreasing the acceptance ratio also gradually reduced the parameter errors (**Supplemental Figure D-F**), as the peak of the posterior distributions better reflected the true values used. There were very small gains by reducing the acceptance threshold below 1%. Further, we investigated how the diffusion estimates would be affected if we introduced a non-zero death rate into the simulation (**Supplemental Figure 7G**). The parameter errors were consistently higher compared with the version with a zero death rate, indicating that it was slightly more difficult to estimate diffusion with a non-zero death rate.*

*For the drug screen experiment, the true parameter values were unknown. Instead, we sought to look at how stable the parameter estimates were while varying the acceptance ratio. If the parameter estimates change drastically for different acceptance thresholds, then the posterior would not be stable and we would need to increase the number of simulations used. We used the parameter estimates using the 0.5% acceptance ratio as a reference and compared how different the estimates would be if we instead had chosen another acceptance ratio (**Supplemental Figure H-I**). The estimates were stable with only minimal changes to the parameter estimates below 10%. Our proposed simulation bank was thus large enough to form consistent parameter estimates."*

We also noted a minor error in the number of simulations created and used. It should have said 100 000 simulations for each set, and 500 closest simulations kept. We have fixed this in the paper as well.

- 2. Related to the previous point, with additional simulations the authors show that allowing a positive death rate ($\mu > 0.0$) does not bias the estimate of the diffusion parameter (Supp Fig 1G). It would also be informative to know whether including a death rate decreases the accuracy of the inferred diffusion rate by plotting the death rate vs the quantiles of the posterior distributions (and ground truth).*

Response. Thank you for the comment. Good intuition. See 1.

- 3. In 'parameter estimation' (in methods): '...we reuse simulations for wells with identical conditions'. Given our comment (1) the authors should show that their bank*

of simulations is large enough for this to be OK (showing posterior convergence would probably suffice).

Response. Thank you for the comment. We have now added Supplementary Figure 7 to document posterior convergence, see above.

- 4. When using their approach to investigate the effects of compounds on cellular migration, there is no discussion on the impact of potential resistance phenotypes. The mode of resistance inheritance could dictate the spatial distribution of cells that remain: for example, if cells rapidly transition in and out of a resistance via a non-genetic mechanism, one might expect resistance to be more spatially diffuse, spread more evenly amongst dividing cells. This could impact the expected spatial distribution of cells post-treatment and hence has the potential to bias the model's diffusion estimates. Whilst the authors choose 'non-cytotoxic' concentrations of compounds, this is only achieved via choosing wells with a sufficiently high proliferation rate – however this is the net rate for the entire well. It is possible that this net proliferation is achieved by a sub-population of resistant cells. Could these issues either be addressed in the model or excluded via analysis of the time-series data during drug exposure?*

Response. Thank you for pointing this out. We agree that it is important to discuss the potential impact of resistance. This has been addressed by adding a paragraph in the discussion (p11, line 412)

- 5. How reliable is the cell segmentation DCNN? If certain compounds influence the morphology of cells, could the networks mistakenly call larger dysmorphological cells as a group of smaller cells? This would interfere with the model inference. Some example data of treated and non-treated cell segmentations would help convince readers that this approach can confidently identify individual cells.*

Response. Thank you for the comment. We added a figure (**Supplementary Figure 2**) showing examples of segmented cells from both the untreated and treated timelapse experiments as well as from the endpoint images from the drug screen.

- 6. How is the radius of a cell calculated, given cells are often not spherical? As the PCF works by drawing a circle of radius r around the centre of the cell, cell morphology could bias the statistic and, by turn, the model. Related to the previous point, could some example estimated radii be shown overlaid on some incucyte images?*

Response. Thank you for the comment. We added the following section to the methods to clarify. "Cell radius was estimated from the images using the segmented images (**Supplemental Figure 2**). The area for each cell was calculated, and the radius calculated by treating the cell as a disk. Note that these cell radius estimates are not needed for the model and were only used to compare the results with the cell radius estimates using the ABC-method." (p14, line 515). To further understand the effect of cell radius, we have added **Supplementary Figure 1C-D** (for simulated data) and **Supplementary Figure 4C-E** (for the drug screen data) to substantiate the estimation of cell radius using our method. We also

added corresponding text to the relevant results sections. Supplementary **Figure 1C** shows the agreement between the observed (cell area) and estimated (circular) cell radius.

7. *We don't think there is a mention of the necessity of deriving a single cell solution at the beginning of the experiment. Presumably this is important, as without it the in vitro experiment would break the model assumption that cells are distributed uniformly throughout the well at $t=0$. Whilst this might seem obvious, deriving single cell solutions can sometimes be challenging; a brief point highlighting this would aid the adoption of this approach by other experimentalists.*

Response. Thank you for the comment. Our primary patient-derived cells are grown as adherent monocultures and are being trypsinized and reseeded as single cells before each experiment. The number of seeded cells to achieve optimal confluency was optimized before the start of experiments of achieving a subconfluent growth phase (approximately 70%) at the end of the assay. Because the cells are in a single-cell suspension upon seeding, the assumption of random seeding should hold. This has now been clarified in the methods section (**p12, line 460**).

8. *As a suggestion for future work, not necessary for this paper, (related to the above) is the method more reliable if before and after images are used (rather than just after images). We can imagine it could help particularly in the setting when proliferation rates are relatively low.*

Response. Thank you for the suggestions. This is a great idea. We now outline a method for including spatial information present in before-images in the Discussion (**p12, line 435**). Its implementation and evaluation are reserved for future work.

9. *In Fig 2, the description of A and B in the legend appear to be the wrong way around.*

Response: Thank you, this has now been corrected.

10. *Some of the units are written incorrectly in the axes labels – for example, in Fig 2, diffusion is given as μ^2 / s – I think this should be $\mu m^2 / s$? (micrometres² per second? Opposed to μ which denotes the death rate in the model).*

Response. Thank you, this typo has now been corrected.

11. *In Fig 2E there are estimates from the time-series data that are orders of magnitude higher than those from the end-point alone images. As the time-series estimates are presumably more accurate, we think these outliers deserve comment.*

Response. Thank you for the comment. Upon closer inspection of figure 2E we discovered that the x-axis was not on a log-scale (as it should have been). After correcting this mistake the number of outliers was reduced, but some still persisted. We believe that heterogeneity in cell motility is a possible reason for the outliers in Figure 2E, and this is now mentioned in the Discussion (**p12, line 420**).

12. *An identity line would be helpful on plots that are comparing true and estimated parameters where the axes limits differ (Fig 2B, for example).*

Response. Thank you, we agree. This has now been added to multiple main and supplementary figures.

Reviewer #2 (Remarks to the Author):

In this manuscript, the authors describe a method to estimate cell viability and migration from single end-point images of patient-derived glioblastoma cells and propose it as a method for standard drug screening. The proposed method offers the possibility to measure cell migration without requiring temporal data (such as, time lapses) or specialised assays (including scratch assay or Transwells-based migration assays). The authors offer the method as an open software. Overall, the manuscript describes an interesting method with a potential application in the study of cell migration, which could be potentially used in standard drug screening. A few general questions arise after reviewing the manuscript.

Response. We would like to thank the reviewer for the valuable comments, which have improved the manuscript.

1. *The authors claimed screening of cell viability, however there are no measurements of viability in this manuscript. Cell death is mentioned, but the only outcomes presented are proliferation and diffusion rates. In fact, it is stated in a part of the study that death rate was not able to be estimated. How does this affect the robustness of the method? Therefore, claims about cell viability should be toned down and implications of not accounting for cell death should be clearly stated.*

Response. Thank you for the comment. We agree that (net) proliferation is the correct term in the absence of direct measurements of cell death. We have adjusted the language accordingly throughout the manuscript. We have now clarified the point that the absence of direct observation of cell death does not affect the estimated of diffusion, nor the estimated effect on net proliferation (**page 6, line 165-178**).

2. *Additionally, the method seems to be limited when cells have a low rate of proliferation. Thus, when screening potent drugs that affect cell proliferation pathways, it may not be possible to use this method to assess these drugs limiting the application to drug screening.*

Response. We fully agree that our method requires some degree of active proliferation to estimate diffusion effects. This topic was covered in the Discussion, (**page 10, line 371**) and also mentioned in Results (**page 5, line 124**). (This restriction would not apply in a version of our assay that would involve paired (before + after) images, as suggested by Reviewer 1, c.f. **page 12, line 435**). In practice (c.f. Figure 4A), many drugs have combined effects on net proliferation and migration, and these are well-suited for investigation with our method.

- 3. In drug screening, when screening a drug in a concentration-response mode, certain parameters, such as IC50 values, are key outcomes. Concentration response curves were plotted in Figure 5 from the data obtained from a time-lapse experiment. However, no data was shown for the end-point analysis using this new method. Measurement outcomes described along the manuscript are proliferation and diffusion rates. How relevant are these in the context of drug screening? Would be relevant to calculate and compare IC50/EC50 values using the end point analysis and time lapse.*

Response. Thank you for the comment. We agree that IC50 are commonly used and that some readers will find it helpful to see such curves to gain intuition. We have therefore added such data as supplementary figures in applicable cases. Also, note that IC50 data is provided for our 94 drug screen in Johansson et al, Cell Reports 2020. As we explain in the introduction of the manuscript, it has high intrinsic value to investigate the effects that drugs have on cells, other than IC50. In particular, diffusion rates are certainly relevant as they reflect the motility of the tumor cells, which is known to be a crucial factor behind tumor invasion. We added IC50 values (on cell count) for the time-lapse experiment as **Supplementary Figure 6B-K** to pair with the dose response curves on migration as was shown in **Figure 5**.

- 4. In the initial part of the manuscript (Figure 1), when validating the method, the authors compared the raw image and the simulation images. In cells with a low diffusion, these two images do not show a clear overlapping with a discrepancy at shorter distances. Based on this, it seems that for cells with a low diffusion, the correlation is not as accurate. Does this affect the overall analysis of proliferation and diffusion? Is the method reliable for such cultures?*

Response. Thank you for the comment. The perceived discrepancy is due to the mathematical model being a simplified representation of the actual behaviour of the cells. All mathematical models of natural systems need to strike a balance between data fit and simplicity ('as simple as possible but not simpler', as physicists say). A key point of our manuscript is that our abstracted model is sufficiently powerful to estimate the net proliferation and diffusion effects of drugs in individual patient-derived cell cultures. In the example mentioned, the exact height of the peak is less meaningful and rather it is the existence of a peak in the radial distribution function, which is relevant.

Minor points:

- 1. Line 76: the results start by addressing Figure 1C with no mention to Figure 1A-B till later in the results section. The images are showing up in order.*

Response. Thank you for spotting this. This has now been addressed, we added references to Figure 1A-B earlier.

- 2. Line 79: Spatial distribution may vary between wells. The same cell line should show similar distribution across the wells unless the seeding was not properly done. One would expect the same distribution across different wells.*

Response. Thank you for spotting this typo. What we meant to say is that the spatial distribution differs between patients, not wells. This is now clarified (**p4, line 80**).

- 3. In general, figure legends are very brief and lack specific details. For example, in Figure 1, low and high diffusion cells correspond to which cell line? No details about the name of the cell line, passage, and other details. Also, no scale bars on any of the images. Same for Figure 4C.*

Response. Thank you for the comment. We have now added additional detail, while conforming with the journal guidelines.

- 4. Line 90: It is stated in the text that cells with low cell diffusion tend to cluster together, while high diffusion does not form clusters. There is no reference that supports this statement.*

Response. Thank you for the comment. Note that this is a statement about how our simulation works, which is a result of this paper. It is quite logical that a simulated population of randomly seeded cells that proliferate without moving will gradually form clusters.

- 5. Figure 5, concentration is represented as $\mu\text{M}/\text{L}$, which is incorrect. Does it mean concentration in μM ? Graphs are not very intuitive. Numbers in the bar graphs correspond to the cell ID?*

Response. Thank you for pointing that out. The drugs are used as μM concentration for dose-dependent treatments, We corrected the concentration units in figure 5.

- 6. Overall, when referring to supplementary data there is no specific indication of the Figure that the text is referring to. If data in supplementary information is cited in the main text, it should be clear to which particular dataset the author is referring.*

Response. Thank you for the comment. Journal guidelines informed us to cite a single (Excel) file of supplementary data. We have, where motivated, added words to indicate what data item to look for in that file.

- 7. The correlation between the transcriptomic profile of the glioblastoma cells and migration response are interesting, however the authors offer a very superficial analysis with no in-depth analysis. Even though the aim of the manuscript it to describe the new analysis method, if the data is introduced, it should present detailed analysis.*

Response. We tend to disagree somewhat with this comment. The key question that we sought to address was if the measured proliferation scores and diffusion rates correlated (or not) with some of the key signatures that very many in the brain tumor field are interested in. We feel that there is enough detail in the manuscript to substantiate this important, specific point.

Reviewer #3 (Remarks to the Author):

The research by Rosen et al describes the development and initial validation of a methodology to estimate cell motility from single end-point images in vitro using a range of GBM cell lines and a panel of inhibitors. I enjoyed reading this manuscript and would like to see it published but have a few comments I would like to see addressed:

Response. We would like to thank the reviewer for the valuable comments, which have improved the manuscript.

- 1. What is the rationale of the choice of inhibitors? There is a mention of 'promising' drugs. Are these clinically relevant drugs as it seems? It would be very interesting to include anti-migratory drugs identified and included in recent GBM studies, such as indirubin derivatives BIO-indirubin or AZD2858; or newly developed drugs such as CCG-1423, alongside the commercial drugs used that had been developed as cytotoxic or anti-proliferative (eg Dasatinib) drugs. It would be interesting to see if there are differences in the effect of the drugs on cell migration using pre-defined anti-migratory drugs.*

Response: We appreciate your comment. The compounds used in the drug-screen were mainly selected from an FDA-approved library. The drug-screen was conducted in three phases, where the number of compounds was reduced in each phase and the number of GBM cell lines was increased, based on changes in viability as measured by an Alamar Blue assay, which is a cell health indicator dye. The selection of promising compounds for the third phase of the screen was also based on the selectivity of drugs indicated by the comparative effect on reference non-brain tumor cell lines. This has been clarified in the Results section (**p6, line 190**). We have now added substantial new experimental results with the proposed anti-migratory drugs BIO-indirubin, AZD2858, and CCG-1423 with a significant reductive effect on migratory potential in one or more of the four cell lines, (**Figure 5A, I-K**).

- 2. Can you comment on observed phenotypes and the effect on cell migration? In figure 4C it looks like the cells are becoming more adherent and sheet-like with increasing drug concentration and may display a switch to a collective type of cell migration? With changes in their migratory behaviour?*

Response: Thank you for the comment. The figure shows cells treated at increasing concentrations of dasatinib. In this case, we find that the cell 'aggregates' were non-migratory, since tracking experiments show a dose-dependent migration-reducing effect (c.f. **Figure 5B**).

- 3. Can you also comment on potential-mesenchymal to amoeboid transition in response to drug treatment and knock on effect on cell migration; this would explain some effects on cell migration opposite to what was expected?*

Response. We appreciate your comment. A paragraph about the impact of different drug concentrations and GBM heterogeneity on drug response has been added to the discussion along with a comparison to previous studies of known anti-migratory drugs in GBM (**p12**). We

agree that the lack of effect on mean square displacement in some of the drug-treated patient-derived cell cultures doesn't mean that the cells are unaffected by the treatment and that other phenotypes such as an amoeboid transition could be detected using alternative image-based assays such as 3D sphere cultures or ex vivo brain slices. We are currently working on such experimental models, however, the mechanism of action of the drugs highlighted as promising here is beyond the scope of this paper.

4. *Can you summarise the findings for the cell lines you used in terms of the subtypes, as these there representatives of all three subtypes and can this information be included in a table?*

Response. The transcriptional subtypes of the PDCs used for the validation studies have been added to Supplemental Data. The subtypes of all PDCs used in the end-point image analysis are listed in \cite{pmid32668248}.

5. *The paragraph in the discussion relating to 3D assays needs expanding (page 10, line 371 onwards). Data generated in 2D may not be representative of data generated in 3D. Have you started to look at analysing data generated, for example, in 3D invasion assays (is this possible in your set-up)? I was thinking particularly with regards to the results from the migration signatures and the transcription factors. Is there more evidence from the literature to support the biological and clinical relevance of these findings?*

Response. Thank you for the comment. We have elaborated on the limitations of 2D models in the Discussion (line NNN). Our migration estimates correlated with migratory signatures and drugs known to be associated with *in vivo* invasion. For example, our top hit, Dasatinib has been shown to reduce GBM invasion in orthotopic xenografts (Huvelde 2013) and was recently suggested to be beneficial to GBM patients stratified based on mesenchymal subtype and high Src activation (Alhalabi 2022). We are - outside the scope of this paper - working with more advanced experimental models such as 3D sphere cultures, ex vivo brain slices, and orthotopic GBM mouse models to follow-up on this and other promising therapeutic options.

6. *When will this application made freely available?*

Response. We will make the application freely accessible on Github. The link has been added to the manuscript (Methods, p18). We will publish a clean working copy of the method before January 28th 2023.

7. *Minor comments: In the result section (first result); I think the mentioning of the figures in text should be in order, so Figure 1A should be mentioned first then 1B etc.*

Response. Thank you for spotting this. It has now been corrected.

8. *Can you check gene names are in italics?*

Response. Thank you for spotting this. It has now been corrected.

References

- 1: Huvelde D, Lewis-Tuffin LJ, Carlson BL, Schroeder MA, Rodriguez F, Giannini C, Galanis E, Sarkaria JN, Anastasiadis PZ. Targeting Src family kinases inhibits bevacizumab-induced glioma cell invasion. *PLoS One*. 2013;8(2):e56505.
- 2: Alhalabi OT, Fletcher MNC, Hielscher T, Kessler T, Lokumcu T, Baumgartner U, Wittmann E, Schlue S, Göttmann M, Rahman S, Hai L, Hansen-Palmus L, Puccio L, Nakano I, Herold-Mende C, Day BW, Wick W, Sahm F, Phillips E, Goidts V. A novel patient stratification strategy to enhance the therapeutic efficacy of dasatinib in glioblastoma. *Neuro Oncol*. 2022 Jan 5;24(1):39-51.
- 3: Johansson P, Krona C, Kundu S, Doroszko M, Baskaran S, Schmidt L, Vinel C, Almstedt E, Elgandy R, Elfineh L, Gallant C, Lundsten S, Ferrer Gago FJ, Hakkarainen A, Sipilä P, Häggblad M, Martens U, Lundgren B, Frigault MM, Lane DP, Swartling FJ, Uhrbom L, Nestor M, Marino S, Nelander S. A Patient-Derived Cell Atlas Informs Precision Targeting of Glioblastoma. *Cell Rep*. 2020 Jul 14;32(2):107897.

REVIEWERS' COMMENTS:

Reviewer #1 (Remarks to the Author):

We continue to enjoy this elegant paper and appreciate the authors' responses to our previous comments.

We accept their comments on the mechanics of the ABC statistical inference, but do think that using a rejection approach (and there are various adaptive methods available) would be better and perhaps a small note could be added into the discussion to highlight this.

The inclusion of the cell death analysis is welcome but the description in the revised manuscript (just Fig S7G?) is overly-brief and little more information is required to make this analysis repeatable.

The cell segmentation data is very impressive!

We look forward to seeing this manuscript in print

Reviewer #3 (Remarks to the Author):

I have reviewed the revised manuscript and the rebuttal letter and am satisfied with the corrections and amendments.